# Current Status of Animal-Assisted Interventions in Scientific Literature: A Critical Comment on Their Internal Validity

**DOI:** 10.3390/ani10060985

**Published:** 2020-06-05

**Authors:** Javier López-Cepero

**Affiliations:** Facultad de Psicología, Universidad de Sevilla, Camilo José Cela s/n, 41010 Sevilla, Spain; jalocebo@us.es; Tel.: +34-954-556-831

**Keywords:** animal-assisted interventions, animal-assisted therapy, human–animal interaction, internal validity

## Abstract

**Simple Summary:**

Animal-assisted interventions (AAIs) have been receiving ever-increasing attention from both practitioners (including psychologists, educators, social workers, and physicians) and clients alike. However, despite this interest, the literature does not provide an unanimous support for including dogs, horses, cats, or other animals in interventions. The present work analyzes whether or not this lack of support could be understood as the result of inconsistencies and/or biases present in the literature, analyzing the definition of AAIs, the role of animals in interventions, the relationship among AAIs and the way humans relate to non-human animals, and the way in which researchers study these phenomena. The present comment provides some clues on how to improve the development of the field, including the following: giving more prominence to cultural, anthrozoological aspects of AAIs; considering AAIs as modalities of well-known interventions, avoiding their representation as “alternative”, “new”, or “groundbreaking”; and making changes to the study and intervention of designs, thus making it easier to demonstrate the impact of human–animal interactions on improving outcomes.

**Abstract:**

Many meta-analyses and systematic reviews have tried to assess the efficacy of animal-assisted interventions (AAIs), reaching inconsistent conclusions. The present work posits a critical exploration of the current literature, using some recent meta-analyses to exemplify the presence of unattended threats. The present comment illustrates that the field (1) comprehends inconsistencies regarding the terms and definitions of AAIs; (2) pays more attention to the characteristics of the animals than to the action mechanisms of AAIs; (3) does not provide a clear connection between anthrozoology (how humans and non-human animals interact in communities), benefits of the human–animal interaction (HAI), and the design of AAIs; and (4) implicitly reinforces these phenomena through research designs. Thus, some conclusions extracted from these meta-analyses need further discussion. Increasing the internal validity of AAIs in empirical studies is an urgent task, which can be addressed by (1) developing a better understanding of how anthrozoology, the HAI, and AAIs relate to each other; (2) highlighting the mechanisms that explain the results in an empirical and specific way; and (3) changing the design of interventions, adopting a component-centered approach, and focusing on the incremental efficacy and efficiency of AAI programs.

## 1. Introduction

Animal-assisted interventions (AAIs) have received increasing attention over the past decades, reflected in a steady growth in their productivity. Previous studies on AAIs show that these interventions are widely accepted by practitioners in different fields, throughout different countries such as the U.S. [1], Australia [2], Spain [3,4], and Norway [5,6]. However, these attitudes seem to be strongly related to personal experience with pets, rather than to knowledge of the technical applications of AAIs [3,5,6]. Many authors have posited the existence of a positive bias that may blur the borders that separate anecdotal and systematic reports in studies that encompass the human–animal bond [7,8], thus threatening the validity of conclusions that could be drawn [9,10,11].

In the scientific literature, meta-analyses are particularly relevant, as they allow us to review the production around a subject matter, drawing a quantitative assessment of the achievements obtained. The literature includes a good number of meta-analyses that measure the efficacy of AAIs (i.e., how well they work), from classic studies [9,11,12] to the most current [13,14,15]. Recent studies show two improvements over older ones. First, studies with a control group, randomised assignment to groups (randomised control trials—RCT) and/or quasi-experimental designs (QE) have become more prevalent, displacing less rigorous, anecdotal reports. Second, an increasing proportion of studies comply with protocols such as the Cochrane Risk of Bias Instrument for RCTs [16], the Methodological Index for Non-Randomised Studies (MINORS) for QE studies [17], or the Preferred Reporting Items for Systematic Reviews and Meta-Analyses (PRISMA) for systematic review studies [18]. These strategies try to improve the internal validity of the studies or, in other words, to guarantee that they measure what they intend to measure (i.e., that the intervention led to outcomes).

However, review studies aimed at measuring the efficacy of AAIs have not reached a single, shared conclusion regarding the usefulness of these interventions. One section of these meta-analyses finds positive results, for instance, Cooke et al. [19] analyzed the effects of dog-training programs carried out in U.S. prisons, finding effects on internalizing behaviors (such as self-esteem, depression, and well-being) and on externalizing behaviors (such as relapse and aggression); Charry-Sánchez et al. [15] reported evidence of pain reduction in children and adolescents under different circumstances (whether associated with surgery or not), and in the gross motor function of children with cerebral palsy; and Virués-Ortega et al. [20] found improvements in social functioning and a decrease in depression, anxiety, and behavioral disorders in the elderly and/or people with mental disorders. However, other studies have found null effects, such as in an analysis of AAIs with dogs in older people with dementia [13], which found a slight improvement in apathy, but null effects in the rest of the variables measured (daily life activities, depression, quality of life, agitation, and cognitive impairment), while another study [14] found improvements in some symptoms associated with dementia (e.g., depression and agitation), but no effect in other areas (e.g., well-being, cognitive functioning, or daily life activities). In this way, after more than a decade, the efficacy of AAIs has still not attracted unanimous support.

Beyond the capability of reaching results (i.e., efficacy), some authors have discussed the pertinence of implementing AAIs given their cost (i.e., efficiency). AAIs imply preparing and keeping of animal subjects [15], adding difficulties in terms of transport, inclusion of a second professional to guide the animal (or, where the work is done by a single professional, the need to divide attention between the client and the animal), and risks for the well-being of non-human animals [21,22,23]. Consequently, any AAI program is more expensive than a parallel intervention that does not include an animal, affecting the eligibility of AAIs [14,24].

## 2. Objectives

In sum, recent meta-analyses offer inconsistent support regarding the use of AAIs. Compared to older meta-analyses [9,11,12], recent works [13,14,15] have devoted greater attention to the quality of the design and to the outcomes obtained, attempting to achieve stronger empirical knowledge, and removing anecdotal information. In spite of this, the present manuscript posits that some key aspects affecting the internal validity of AAIs remain neglected in the current literature, as exemplified by recent meta-analyses. The first aim of this work encompasses a critical exploration of the current threats that AAIs face, including (a) common inconsistences regarding terms, definitions, and classifications; (b) the confusion around the role animals play within AAIs; (c) the absence of explicit mechanisms of change; and (d) the implications that various research designs have on the conclusions drawn.

The second aim relies on discussing the guidelines to overcome these risks, including (a) the use of definitions and classifications that are coherent with pre-existing disciplines; (b) to pay less attention to what the animal does, and more to the mechanism or strategy of change; (c) to use a wider approach, in order to connect AAIs, the benefits of the human–animal interaction (HAI), and the cultural context; and (d) to adopt a component-centered approach, which would booster research in the efficacy and efficiency of AAIs.

## 3. Analyzing Key Points That Affect the Internal Validity of AAIs

### 3.1. Classical Terms and Definitions of AAIs

There is a wide range of terms that combine different forms of intervention (therapy, education, activities, etc.), support functions (assisted, facilitated, etc.), and references to non-human animals (pets, companion animals, or diverse species such as dogs or horses) [25,26]. From all of the possible combinations, the most commonly used label in the field is animal-assisted therapy (AAT; currently included as a descriptor in the thesaurus of PsycINFO and Medline), followed by animal-assisted activities (AAAs) and animal-assisted education (AAE).

The first mainstream definition of AAIs was offered by the Delta Society [27], which differentiated AATs as programmed and evaluated procedures, in contrast to AAAs, defined as playful and spontaneous (AAE was initially included within AAAs). Although a later publication clarified that AATs should be carried out by qualified health professionals [28], the original classification of the Delta Society remains widely used, even today. The International Association of Human–Animal Interaction Organizations (IAHAIO), which brings together some of the main organizations dedicated to AAIs worldwide, has maintained the difference between AAT (purposefully designed, with specific objectives and evaluated) and AAA (spontaneous, informal) in its guidelines [29], thus fostering the association between any “designed intervention” and AAT.

Some reviews have pointed to the generalization of the term AAT as a standard to refer to any form of intervention that includes animals, even for programs that would hardly be qualified as psychotherapy, physiotherapy, occupational therapy, or any other therapeutic process [26]. A search in databases such as PsycINFO shows that not all publications classified under the main subject AAT include any derivative of the word therapy as a keyword, while other labels, e.g., education, are used marginally [26].

Some classic works highlighted these difficulties explicitly, expressing the impossibility of differentiating AAA from AAT in their analysis due to insufficient definitions [10,12]. However, the inconsistent use of descriptors, and the discrepancies between the use of the term therapy (generic in AAIs, restricted in disciplines such as psychology, occupational therapy, and physiotherapy) still remain, highlighting the risk of confusion in this field. Confusion represents a direct threat to internal validity, as it is impossible to determine that an intervention works when researchers and practitioners do not have a clear view of which characteristics define it.

The present work uses the term AAI, understood as any intervention that includes a human–animal interaction (HAI), as an integral part of its design (for a complete definition, see below).

### 3.2. Empirical Definition of AAI Studies

Quantifying the presence of confusion regarding the terminology and definitions of the entire area would be a complex task, exceeding the aims of present work. However, the existence of these inconsistencies within recent meta-analyses is indicative that this confusion is common, since the meta-analyses aspire to compile the most rigorous works on the subject.

It is interesting to identify the criteria those works use to include and exclude studies in their analysis, as they provide an implicit, empirical definition of what AAIs are (and what they are not). Surprisingly, not every meta-analysis provided an explicit definition of the subject [15]. Some recent studies analyzed together the results obtained from programmed assisted interventions and those experiences in which the interaction is playful and spontaneous (AATs and AAAs, according to usual definitions). A paradigmatic example is that of Souter and Miller [12], who chose to mix AAA and AAT due to the difficulty in differentiating between both constructs. In their review, Charry-Sánchez et al. [15] incorporated works that took from 10 min in a single session to 60–70 min per week in long-term programs, labeled indistinctly as AAT. Virués-Ortega et al. [20] jointly assessed the effect of visiting programs—in which the interaction is free—with regulated intervention programs, such as therapies. Although not exhaustive, these examples show that AAA and AAT may be indistinguishable in some of the available meta-analyses. Finally, other works have gone further, including results associated with cohabitation (that is, without intervention) in their analyses [9,30,31].

In sum, reading recent meta-analyses helps us to exemplify inconsistences in the field. Trying to measure the efficacy of AAIs by gathering studies about effects of the HAI in very different formats (cohabitation, occasional contact, regular visits of the animal, etc.) and with different objectives (from therapy to companionship) is a clear example of “mixing apples and oranges”. Although this problem is not specific to AAIs [32,33], reporting a single conclusion for such different events is not appropriate, no matter how well designed the analyzed studies are. Therefore, these inconsistences represent a clear threat to the validity of any conclusions that those studies draw.

### 3.3. The Role of Animals within the Intervention Strategy

Each of the reviews and meta-analyses previously cited determines, within its methods, the requirements regarding sample size, the quality of the intervention design (the use of control groups, randomized assignations, etc.), or the frequency and intensity of contact needed to be included in the studies. However, as stated before, these works analyze together studies that have little in common, with the animal’s presence being the main shared criterion for inclusion. That is to say, the reasons for the grouping seem to correspond more to esthetic questions (e.g., the presence of the animal) than to explanatory ones (e.g., why this presence helps to achieve outcomes).

Publications regarding AAIs tend to offer detailed information on the physical characteristics of the animals (breed, color, sex, neutered or not, etc.) and their training and/or certification. However, unlike service animals (e.g., guide dogs), the intervention animals always work under the supervision of the guide who accompanies them—a measure that guarantees the well-being of human and non-human participants, as proposed by the OneHealth framework [22,34,35]. Therefore, animals do not need to make decisions autonomously (such as crossing a road or avoiding an obstacle).

A complete description of the animal adds more detail to the method but does not provide information on the role, or the functional value, that the animal played in the program. As early as 2007, Nimer and Lundhal [11] highlighted the scarcity of information about the treatment plan in most of the studies that they reviewed. Several recent meta-analyses codify and describe the animals’ role through the actions they perform (playing, grooming, walking, etc.), without explaining how these activities could support the intervention strategy. In the 10 studies analyzed by Zafra-Tanaka et al. [14], only three reported an action identifiable as a professional role or strategy—cognitive stimulation, improving fine motor skills, etc.—while six referred to specific activities—petting, feeding, walking, etc.—and one did not provide any information at all in that respect. In their meta-analysis, Hu et al. [13] described the objectives of every study (e.g., increasing physical activity or promoting communication) and the activities carried out with the animal, without referring to the underlying strategy or change mechanisms that would connect activities to outcomes. In their study, Wilkie et al. [36] provided five definitions of AAI that include horses, and then go on to say that *“for the sake of simplicity (...) any reference to an intervention involving a horse has been referred to as equine therapy”* [36] (p. 379), carrying out a single analysis with works that referred to AAT, AAA, AAE, equine-facilitated learning, and hippotherapy (i.e., physical rehabilitation [25,36,37]).

The above-cited meta-analyses exemplify that, more than a decade later, the literature keeps paying more attention to the presence of animals by themselves than to intervention strategies, thus preventing replicability of the studies—one of the main conditions to achieve scientific knowledge. This is especially striking, given that recent meta-analyses have tried to increase internal validity by including studies that meet the highest standards and protocols in order to control the risk of bias. Given this, the next question is: how can we connect the presence of animals to intervention outcomes?

### 3.4. Mechanisms for Explaining Changes

Many definitions characterize AAIs by their expected effects. Without being exhaustive, the reader may find proposals such as, “*[AAI] uses a variety of animals (e.g., horses, domestic pets, farm animals) to achieve psychological benefits in a wide range of populations”* [37] (p. 23); “*AAT is the deliberate inclusion of an animal in a treatment plan (...) to achieve predefined outcomes believed to be difficult to achieve otherwise or better outcomes addressed to animal exposure”* [11] (p. 225); or “*AAT are interventions in which animals participate as an integral part to improve specific outcomes in the patient (…) This interaction may include various activities such as petting, brushing (...)”* [14] (p. 1). However, few works have focused on the mechanisms that lead to these benefits.

A myriad of studies assess the effects of the human–animal interaction (HAI) on physical health, psychological well-being, and integration into the community, among others [21]. Most of the reported effects are positive, probably because of a selection bias—studies usually gather participants that choose to cohabitate with animals. However, our interactions with non-humans can also affect us negatively because of economic costs, injuries, zoonosis, etc.; thus, the size and universality of these effects remain under discussion [8,21,22].

AAIs include the HAI within intervention programs, and thus it is relevant to analyze the mechanisms available for explaining benefits. We can analyze the potential effects of the HAI on human health by using three basic conceptual schemes: direct effects (e.g., activating physiological processes that affect health, such as reducing stress by classical conditioning), indirect effects (e.g., activating behavioral patterns that facilitate health, such as dog-walking), and those explained by a simple correlation without any causal link (e.g., when a very active person chooses to live with a dog and to play sports) [38].

In any case, the HAI and AAIs are different subjects of study, and should not appear as similar or interchangeable concepts. Beyond this, practitioners should not assume that the benefits associated with the human–animal bond will be replicated within intervention programs—e.g., animals may act as distractors, making it difficult to carry out some techniques. In sum, when practitioners and researchers include non-human animals into their intervention programs, they should be able to explain how the HAI could help to reach their objectives.

The relationship between the HAI and the intervention remains unexplored in most studies, although the literature provides some good examples. First, Virués-Ortega et al. [20] attributed the positive effect of HAI to animals’ stress-buffering value—explained by a combination of perceived social support, classic conditioning, and social facilitation—and because it triggers healthy behavior, e.g., going for a walk. Secondly, an experience carried out with a control group (that received the same treatment without animals) evaluated the differential impact of a dog’s presence in the therapeutic alliance, a variable related to intervention success [39]. Explicit mention to the mechanism of change helps to discuss outcomes, and makes it easier to study intervention strategies in a systematic way.

### 3.5. Do Animal-Assisted Interventions Exist?

Specific mention of the mechanisms activated when including animals in AAIs brings us to a relevant question: are AAIs a form of independent intervention, like psychotherapy or speech therapy?

Some authors define AAIs as differentiated from other interventions, either as a “*viable alternative to conventional intervention strategies*” [36] (p. 377) or like “*an adjunct to other interventions*” [11] (p. 235). However, other authors consider that the presence of the animal is an additive element in a treatment plan. That is, the impact of the interaction between the client and the animal should be evaluated to determine its incremental value on the change process, treating it as a concrete component within the program [10,21,40]. From this “component-centered approach”, AAIs are not considered as separate interventions, but as modalities—that is, modifications—of pre-existing interventions rooted in disciplines such as psychology, education, and social work, among others.

The designs used in each study imply different ways of representing AAIs. As an example, in the aforementioned study, when the same intervention program was carried out in two groups in parallel, with the exception of one specific component—with and without a dog [39]—the researchers reflected on the fact that the HAI is a component that may or may not be included. Comparing the outcomes of both groups allows us to determine how much difference the HAI can make within the treatment process. On the other hand, it is easy to find studies in which one group receives AAIs while the control group remains without treatment (known as the passive control group, i.e., waiting lists). Comparing an AAI psychotherapy program to no intervention allows us to measure the impact of the intervention as a whole package, but does not provide any information on how much of this change is attributed to psychotherapy, to the HAI, or to their interaction.

However, recent meta-analyses do not address the impact of these design differences. Maujean et al. [37] included two studies with active control groups (the same treatment without the animal) and five studies developed with passive control groups (no treatment). Cooke et al. [19] compiled studies without a control group, with active control groups, and with passive control groups. Furthermore, Zafra-Tanaka et al. highlighted the “*heterogeneity of the control group intervention”* [14] (p. 7) as a methodological limitation, but still drew conclusions from the 10 studies as a whole. Other meta-analyses have neither indicated nor discussed the importance of this factor [13]. Thus, it is hard to determine if the conclusions drawn from those studies reflect the efficacy of whole programs, or the relative impact of the HAI within a given intervention.

Designs including passive control groups (without treatment) reinforce the image of AAIs as a separate field, different from broader disciplines. Many authors have discussed the importance of turning toward a component-centered approach, as it would help to depict AAIs within the scientific and professional logic of disciplines such as psychology, occupational therapy, social work, and education [10,21,40]. In this regard, active control groups (parallel in everything except for one specific component, e.g., with and without the animal) are of help in learning how much the HAI improves an intervention, enabling the conclusions reached to be integrated into the corpus of knowledge of the disciplines that they are rooted in (Figure 1).

## 4. Proposals for Integrating AAIs

### 4.1. Improving the Definition, Classification, and Terms Used

AAIs consist of incorporating an animal into an intervention program. More specifically, they involve some form of a HAI. Given than AAIs do not represent a different form of intervention, nor an alternative one, we do not need to establish a specific classification for AAIs, since an intervention is (or not) a therapy, education, or play activity, regardless of whether an animal participates. Riding a horse or playing with a dog does not automatically turn a program into an AAT or an AAE. Practitioners should select terms reflecting this idea, keeping coherent with the regular use in main disciplines.

The AAI label is an umbrella term capable of grouping all of the works that take advantage of the HAI as an enhancing element. Likewise, mentioning the animal (cat, dog, etc.) will make sense when the species makes a difference (as is the case of a horse in physical rehabilitation). When the effects are not directly concretely tied to the animal—such as when the aim is to create warm emotional bonds, perceived social support, or opportunities to establish social contact with third parties—using the generic term “animals” avoids unnecessarily atomizing the terminology, thereby maintaining a greater consistency in the field.

These aspects are included in the following definition, adapted from López-Cepero [21],
“Animal-assisted interventions offer a generic label to bring together any programmed intervention (designed and evaluated) that takes advantage of the benefits of human–animal interaction as an enhancing or facilitating element. AAIs do not in themselves constitute a profession or discipline, but rather they are identified by including animals as support for any pre-existing professional role, such as performing psychotherapy, physiotherapy, education or even sociocultural animation; therefore, the professionals involved must comply with the legal and qualification requirements needed to perform these tasks. In all cases, the design of the intervention will assess the relevance, efficiency and ethical implications of including animals within the given anthrozoological and sociocultural context. The training and certification of the animals will depend on the tasks they are entrusted with, but they will always act under the supervision of qualified personnel to guarantee their well-being and the safety of all participants.”

The field of AAIs involves many organizations, practitioners from a variety of disciplines, and researchers. Thereby, communication represents a challenge. It is important to choose the simplest and widest terms and definitions available, and to use options that are compatible to those of use in every actor’s discipline. Creating new terms or definitions is only justifiable when the work done is qualitatively different from other pre-existing fields.

### 4.2. Attending to Mechanisms of Change: How the HAI Can Improve Intervention Programs

Practitioners and researchers should include an explicit mention of how the presence of the animal may enhance interventions, matching the benefits of the HAI with key strategies and mechanisms described in their discipline frameworks. This description should provide information of the strategic, operational value of the presence of the animals—not only of their characteristics—within recognizable interventions and/or techniques, thus allowing replication of the study. The following paragraphs provide some illustrative examples, using a model that integrates some common factors, i.e., contents and processes that are present in all approaches.

Wampold’s Contextual Model [41] offers a framework to better understand how different psychotherapies work, although it can be easily translated to other disciplines that try to change human behavior. This model establishes that three common factors are present in all interventions: (a) they involve interpersonal relationships, (b) they include expectations about the process of change, and (c) they imply specific techniques.

Without being exhaustive, we can provide some examples for each of those factors. First, it is possible to take advantage of the effects that the HAI has on the clients’ perception of safety and positive acceptance, in order to establish a better work environment [42,43], or to evaluate its effect on their commitment to the process of change (within the so-called therapeutic alliance) [39]. Second, as far as the clients’ expectations are concerned, the literature shows that the presence of animals can affect their perception of the therapist and their attitude toward disclosing personal information—elements that affect the probability of success [44,45]. Finally, the acute effects of the HAI can enhance relaxation (which is useful in exposure techniques such as systematic desensitization), serve as a distractor (something used in pain-management programs), serve as a behavioral model (at the workplace or in the classroom), support the performance of role-playing, etc., thus supporting the implementation of specific techniques.

In this way, the inclusion of animals in these three areas—relationships, expectations, and techniques—enhances the existing elements to accelerate (quantitatively) the process of change, acting through direct (e.g., classical conditioning, perceived safety, reinforcing value) and indirect (e.g., activation of healthy behaviors such as walking or playing) mechanisms [20,21,38,46]. Some interventions aim to achieve acute impacts (e.g., visiting programs in pediatric wards), thus the spontaneous effects of the HAI can be used to justify the presence of the animal. However, if the aim is to allow lasting change (as in therapeutic or educational programs), practitioners should give a clear explanation of why, and under what conditions, the presence of the animal could help to achieve medium- and long-term objectives. In this way, the inclusion of animals is justified through mechanisms that we can hypothesize and verify using scientific methodology, ruling out any magical hypothesis in this regard.

### 4.3. Understanding AAIs from a Cultural Point of View

The effects of the HAI are not universal, nor can they be presented as a panacea [7]. Not everyone experiences the same phenomena when in contact with other species—as it depends on variables such as their personality, experience with animals, etc. [22]—and not all cultural contexts make each one of these effects equally likely. Some of the effects derived from the HAI work through direct pathways (e.g., reduction of stress by classical conditioning), while others do so in a mediated way (e.g., improvement of cardiovascular health through dog-walking [38]), but all of them occur within a wider context that makes certain narratives, values, and habits more probable [21,42].

Just as happens in education, psychotherapy, and other practices that aim to modify human behavior, practitioners must attend to the cultural fit of the techniques they implement. Animals play a role in the community, affecting and being affected by humans. Learning about these interactions, symbolic values, and social roles—studied under the label anthrozoology—would help to establish a catalogue of opportunities for professionals to have at their disposal [8,47]. As a corollary of this, practitioners should attend to meanings and habits derived from their patients’ cultural background in order to raise the chances of success.

### 4.4. Improving Research on the Efficacy and Efficiency of AAIs

Adopting a component-centered perspective (i.e., understanding AAIs as being a mix of a HAI and a pre-existing intervention program) may help to obtain greater conceptual clarity in the field, with immediate implications for both research and practice.

First, it would allow an easier development of studies regarding efficacy (i.e., how well AAIs work). Instead of trying to demonstrate that AAIs could work as a separate field, researchers could use the knowledge already amassed in evidence-based disciplines by selecting intervention programs with empirical support, and by assessing the relative impact of the HAI on the final outcomes. Establishing the differential impact of the animal’s presence in a program that is already validated is noticeably simpler than starting the process all over again.

Second, considering the HAI as a component makes it easier to analyze its efficiency by comparing the costs and benefits offered by two parallel programs. Given that the inclusion of a non-human animal implies higher costs—including transport, preparation, and ethical concerns, among others—it is important that practitioners are ready to explain in which ways and at which stages the HAI will be useful. As stated before, the presence of animals affects different elements of the intervention (e.g., boosting the therapeutic alliance, creating better expectations, and increasing the impact of specific techniques), and thus there is a strong rationale to justify the incorporation of the HAI. However, at the same time, it would be of interest to assess the differential impact of the HAI on every factor, given that some clients may enjoy the direct interaction with animals (e.g., fostering a warmer, closer relationship) and, at the same time, feel it is unprofessional (e.g., worsening expectations toward the practitioner’s training), thus having mixed effects. On the other hand, it is necessary to assess if the presence of an animal may be beneficial at some stages of the process, while negative at others (e.g., an intense HAI may help to create a stronger relationship at the beginning of the process, but may act as a distractor in the middle stages).

Thus far, the literature offers little information in this regard. To face this challenge, research on efficacy and efficiency should involve the use of active control groups (e.g., groups that are subject to parallel intervention programs with and without animals [39]), rather than passive control groups (e.g., one group receive AAI while the other receive no treatment, such as in waiting lists). AAIs are not in competition with “no treatment” situations, but rather with other available interventions, programs, techniques, etc., and the design of the intervention must reflect this fact. Given that the literature shows an increasing interest in including the best designs (such as QE and RCT studies), this change can take place in a short space of time.

Third, a component-based perspective makes it easier to incorporate new findings into the common corpus of knowledge of a discipline. Practitioners that are not specialized in AAIs would find it easier to learn, understand, and incorporate the HAI as an additive component. In addition, clients would have a clearer view of what AAIs are, preventing fraud due to professional encroachment—something that happens when someone offers “therapy” or “education” without being legally qualified to do so. In sum, it would have a positive impact on the image of AAIs among researchers, practitioners, and clients alike.

Finally, we must insist that the mechanisms of change implied in the intervention program need to be explicit. The HAI can enhance the intervention in different ways, but it is the responsibility of the practitioner to explain what these ways would be, and to measure whether there are any causal or moderating effects. In this way, we can enhance the image of AAIs as evidence-based interventions. At the same time, it seems illogical to ascertain whether improvements that were not programmed at the beginning, or that cannot be explained (e.g., increased self-esteem derived from physical treatments, or any other improvement achieved in a non-specific manner), are a result of the intervention program. Putting animals forward as a panacea enhances the fanciful view of AAIs.

## 5. Conclusions

The literature on AAIs has grown steadily in terms of both quantity and quality over the last decade. Recent works reflect a greater concern for guaranteeing the validity of studies, as researchers implement the strictest methodological criteria to separate anecdotal information from scientific knowledge [16,17,18]. However, despite the evident improvements, these changes have not managed to solve some of the internal validity problems highlighted in classic works [11,12].

AAIs are widely accepted by professionals and the general population [1,2,3,4,5]. However, the inconsistencies present in a sizeable number of elements—such as their definition, classification, explanatory mechanisms, animal roles, and design of validation studies—are evident. In order to solve this problem, all actors involved (researchers, practitioners, and organizations) need to address this challenge together, as the current situation prevents to obtain a clear view of AAIs efficacy and efficiency, and to provide a clear communication to potential clients.

For this reason, this manuscript proposes future guidelines that aim to provide a unified conceptual framework for the different forms of AAI that we find in the literature. AAIs must grow under the protection of evidence-based disciplines, adopting coherent terms and definitions. In turn, studies focusing on AAIs should give prominence to the additive benefits derived from the HAI and the mechanisms by which they occur. At the same time, the cultural context should be attended to, in order to integrate the different levels of analysis that are found in the literature (anthrozoology—HAI benefits—AAIs). Finally, it is especially relevant to improve the coherence between the means and the ends sought through the intervention, in order to understand how, how much, and under what circumstances AAI programs work.

The field of AAIs is in a difficult situation, in which methodological improvements co-exist with conceptual inconsistencies. Solving these threats to internal validity is a necessary step to organize their growth, to integrate them within broader disciplines, and to protect clients from professional encroachment, fanciful hypotheses, and ideas not supported by evidence.

## Figures and Tables

**Figure 1 animals-10-00985-f001:**
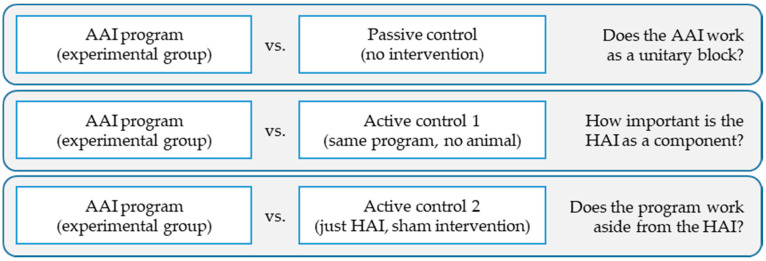
Utility of different control groups (adapted from López-Cepero [21]). AAI = Animal-assisted intervention. HAI = Human-animal interaction

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
