# Peer review of "Current Status of Animal-Assisted Interventions in Scientific Literature: A Critical Comment on Their Internal Validity"

_animals, 2020, doi:10.3390/ani10060985_

Round 1

Reviewer 1 Report

Over the past several years there have been several meta analysis papers on AAI and HAI. Although I believe that they can be extremely helpful in providing insights to the challenges in the field, many including this reviewer believe that a meta analysis paper should also make substantial comments about what the research is suggesting needs to be done. 

That being said, I feel the author makes some viable interesting points. I suggest to enhance the article that the conclusion and discussion area should be expanded to discuss in more detail:

  1. Suggestions on addressing the mechanisms that are impacted by AAI (or not impacted) and how and what researchers need to study and investigate.
  2. Provide more suggestions on the common terminology that the author points out as being a problem. This reviewer believes that the terminology is not as significant a problem, but does agree that at times some researcher don't capture the specifics of the type of AAI in their studies. What suggestions would you give future researchers,  making sure that this issues is dealt with.
  3. 3.On line 349 the writer discuss the need for RCT and I would agree but would point out that there are more studies now attempting to have more rigorous research standards. Perhaps there should be given more attention to this issue and report some of the best practice studies that are there.
  4. What suggestions would you give researchers to clarify the actual intervention that was integrated. At times this void of information impacts the proper replication of the research implemented. Perhaps identify some studies that have done this well.

Author Response

Dear reviewer, 

Thank you so much for your comments. I prepared a detailed response in the attached file.

Kind regards, 

Reviewer 2 Report

Author’s statements in the first paragraph (manuscript lines 42-45) of the Introduction are inaccurate and misleading with the sources cited being severely inadequate representations of AAI studies completed in the countries listed and also omission of many studies from other countries.

Clarification should be made (in lines 51-53) that cited sources 16-18 are not about AAI, but instead about methodology that can (should?) be applied to AAI research.

I applaud the authors underlying reasoning for this work (for example lines 202-204), that practitioners and researchers should be able to explain how human-animal interaction could help reach their objectives.

I strongly agree with the author’s overriding view that activities such as merely riding a horse or playing with a dog does not automatically make an HAI clinically efficacious. This is an important consideration. And, the author’s calling upon each profession (psychology, etc.) to more carefully explain how HAIs work as AAIs with relative therapeutic components is vital for the broader field of AAI to gain greater acceptance.

The author’s attendance to existing organizational operational definitions of AAI applications is excellent, i.e. IAHAIO.

The author’s recommendations for improving methodological designs for AAIs are very useful.

The author’s attention to cultural indicators effecting AAI outcomes is very important.

It is much appreciated that the author emphasizes that AAI is not a separate field, but rather AAI is a tool (modality) incorporated into professional fields and should be described as efficacious relative to therapeutic goals and interventions form those individual fields. A component approach in describing AAIs from the unique perspective of each professional field is a useful way of bringing about consistency and understanding of AAI utilization within each professional field. And, a good way of minimizing confusion and contamination by inadequate and inappropriate AAI applications.

The author does a very nice job of achieving the stated purposes of the manuscript.

This ideas presented in the manuscript are a significant contribution to AAI literature and should be published.

Author Response

(The authors gave the same response as above.)

Reviewer 3 Report

The article was clearly structured and  cogently argued. I found it  an interesting paper to read and it makes a compelling argument for a component based perspective to be adopted. The writers highlighted the need for conceptual clarity ( a point which is echoed in many other papers in the field); and provided a sound overview in the early part of the article on the different ways in which interventions involving animals are defined. It was good (and necessary in light of the arguments being made!) to  set out a definition for the purpose of the paper.

Two points struck me on reading which I would suggest addressing.

1) The abstract describes the paper as adopting "a critical approach to the literature". In some disciplines a critical approach is a broad approach - which involves a critique of the more positivist approaches. Critical Psychology would be an example of this. To avoid any confusion or misinterpretation that the writer is writing from this perspective (which they do not appear to be doing so), I suggest re-wording that statement - perhaps describe it as an evaluation of the current literature, using some recent meta-analysis etc  or this paper will critique the literature, using some recent meta-analysis etc

2) It is an issue that in many papers that terminology is used incorrectly, and agree that universal guidelines and terminology is required. However, in my experience of conducting research in this area, it can be the site at which the animal programme is taking place that has labelled it so, leaving the researcher with a dilemma when writing about the programme. It may be that a therapeutic 'intervention' has even been funded as AAT (though would not fit the definition). Applied research is complex and perhaps a sentence or two acknowledging this would be useful. 

Overall, I think this will be a useful resource for researchers and practitioners, as well as representing a point for discussion of how to improve the quality of studies in the field. I recommend publication

Author Response

Dear reviewer, 

Thank you so much for your comments. I prepared a detailed response in the attached file.

Kind regards, 

This manuscript is a resubmission of an earlier submission. The following is a list of the peer review reports and author responses from that submission.

Round 1

Reviewer 1 Report

Comments, Manuscript 752528: “Current status of animal-assisted interventions in scientific literature: a comment on its internal validity”

The topic that this paper seeks to address is timely and important. That said, I believe that the current manuscript needs substantial revision before it is fit for publication in a journal with as broad an audience as has Animals.

One very problematic element of the current manuscript is the command of English displayed. The paper was extremely challenging to try to understand; the use of English grammar and word choices greatly compromised this native English reader’s comprehension of the author’s meaning. A very strong editing hand is needed here to make the English understandable. Right now, the language presentation is sufficiently poor to challenge the entire value of the manuscript.

The second overall comment I have to make is that the author appears to assume a depth of understanding of his field of study in his readers that I fear is just not there.   The journal Animals has a broad audience and entertains a broad range of topics. Thus it can readily be assumed that professionals from a broad variety of backgrounds comprise its readership. Many (not just those who are themselves involved in animal-assisted therapeutics) would be interested in reading a paper with this title. But many will also be unfamiliar with concepts or constructs such as

Internal validity, randomized control trials, quasi-experimental design, Cochrane’s Risk of Bias Instrument, the Methodological Index for Non-Randomized Studies, the Preferred Reporting Items for Systematic Reviews and Meta-Analyses, or even the concept of a meta-analysis! The author needs to briefly define or describe these constructs, or point to where more may be learned about them, in order to facilitate reader understanding of the manuscript.

Similarly, the author needs to clearly and specifically articulate HOW lack of control groups impacts or otherwise compromises our understanding of the internal validity of AAIs, rather than simply asserting that it is so.

Some specific comments follow, in no particular order:

1) It would be very helpful to readers who are unfamiliar with the construct of “internal validity” to begin by defining internal validity and why it is important.

2) The author should explain in the text what is in Fig 1, and where these data came from or how they were generated.

3) Lines 79-80 I think I know what the author is trying to say here, but this does not say it

4) Lines 91-93: What is the author trying to say here? What is the literature to which he is referring? Is this what is meant?

“The literature on AAI’s brings together a wide range of terms or labels that combine different intervention contexts in which AAI has been used (e.g., physical and/or psychological therapy, facilitation of education), the support function served by the AAI (e.g., assistive, facilitative), and the particular non-human species employed (e.g., dogs, horses).”

As can be seen in my suggested revision, MUCH more clarity needs to be in place in the writing, so that the reader can actually follow what the author is trying to say whithout having to be inside the author’s head!

5) Lines 97-105: As is the case elsewhere in the manuscript, I find these lines totally confusing. “Popularity” is not the same as “frequency of use.” The word “popularity” implies a preference, whereas it may be that these terms are what are currently available for use and are most readily understood by the authors using them, and not necessarily the terms authors would use if an alternative was made known to them—hence the frequency of their use. Does the author mean this?

“These terms are used frequently in the literature, with infrequent attempts to operationally define them. One of the reasons for this may be because of conceptual definitions of the terms initially put forward by the Delta Society, which differentiated AAT’s (the Society originally included AAE’s under this term) as programmed and evaluated procedures, and AAA’s in contrast as playful and spontaneous interactions.”

Again, there are some assumptions in the current style of writing in this manuscript that the reader understands what is in the author’s mind, rather than effort made to clearly and explicitly articulate what the author intends to say.

6) As I am sure the author knows from his detailed study of meta-analyses, it is customary to report the years searched in an assessed literature search. In Table 1, what were the years searched?

7) Lines 117-119: is “validity” the right word here or should it be “ambiguity?”

8) Lines 141-143: I cannot understand this very well. Is this what is meant?

“On the other hand, other works have gone further, and include results correlated with living with animals (for instance, having a pet). Virués-Ortega et al. jointly assess the effects of programming that includes visiting animals (in which interaction with the animal is a matter of a client’s free choice) along with more regulated interaction programs (such as formal animal-assisted therapies).”

9) The author himself defines AAI and HAI differently on page 4 from what is asserted elsewhere in this paper. Yet variability in the definitions of these terms is one of the primary concerns raised in the current paper.

10). Lines 152-157: Support this assertion with citations. Same with Lines 158-163.

11) Lines 165-166: One can observe a reliable correlation between conditions without needing to know (or being able to know) the underlying reasons for the observed correlation. Thus, it is not a necessarily robust criticism of a correlational study (such as a meta-analysis) to say that it failed to “explain how these activities can support an intervention program.” Such explanations should be the purview of the original empirical manuscripts that are included in the meta-analyses, and while it would be appropriate for the authors of such meta-analyses to include some commentary as to the putative mechanisms underlying found correlations, it is not obligatory that they do so.

12) Lines 179-18: Justify saying that the term “hippotherapy” is used only for physiotherapies involving horses; I have seen it used to refer to all manner of HAI involving horses.

13) Lines 184-185: I don’t really understand what the author is trying to say here. Same with Lines 208-210. What does this mean? “Secondly, an experience carried out with a control group evaluate the impact of a dog’s presence in the therapeutic alliance, which is a variable that measures the results of the intervention.” Which noun in this sentence is the “variable that measures the results of the intervention?” And what dog are we talking about? Is this meant to be some kind of example?

14) Between the paragraph ending with Line 211 and the new paragraph beginning in Line 212, there is a substantive change in subject, without any preliminary transitional text.

15) Line 224: “   the impact of the interaction between client and animal can be evaluated….” Would this be better stated as “should?”

16) Lines 231-232: what is meant by an “active control group” compared to a “passive control group?” The author needs to define that briefly for readers that may not understand these constructs.

17) Fig 2: English does not use inverted ? before a question

18) Line 256 makes no sense in English

19) Lines 280-289: who wrote this? Source? Writing style is notably different from the rest of the text. Is this a suggested definition? If so, by whom?

20) Line 293: “i.e” is short for “id est,” and essentially means “in other words.” Does the writer actually mean “eg” here (“for example?”) Might want to include animal wellbeing as another potential negative aspect of AAI’s

21) Line 299: What is mean by “paradigm “ here? The author is assuming more understanding in his/her readers than is likely the case.

22) Lines 300-308: what does this mean? And how does whatever the author is talking about exemplify the “3 aspects” of the model just described?

23) Line 309: what “three areas?”

24) I don’t find fig 3 useful or necessary.

25) Line 336: What is meant here by “component-centered perspective?” Remind the reader! What components do you mean?

26) Lines 338-341: Clearest statement in this manuscript thus far, and important! The rest of the manuscript should be more like this!

27) Lines 346-348: “less interesting?” What is the author trying to say here? Does he mean “less valuable?”

28) Line 351: What is meant by “prevailing research logic?”

29) Line 373: The current manuscript is NOT so much a “study” as it is an opinion paper. If the author wishes to make more of his own casual (as he himself describes it) search of the literature, he should review and summarize those data here.

30) Linr 391: Revise to be more like “in which AAI is just one element of a host of others within the therapeutic environment” or something like this. Need to remind the reader what you mean!

31) Paper shifts quite dramatically in tone, clarity of writing, and English readability from about Line 279 to the end.

Author Response

With kind regards, 

Reviewer 2 Report

I was looking forward to reading this manuscript.  Unfortunately, the writing is sufficiently problematic that I do not think it is suitable for review as written. The awkward wording makes it difficult to understand the author's argument.  Because of these issues, I do not feel comfortable writing a review of the manuscript.

Author Response

Dear reviewer,

Thank you so much for your feedback. I read it carefully and introduced many changes. I can summarize main changes in manuscript:

  1. I rewrote it to adapt the style to an opinion manuscript. Previous version resembled a systematic review in some aspects, and this was misleading.
  2. Manuscript is now 600 words shorter. I removed some paragraphs in order to make the text more direct, and all sections end now with a summary or recommendation.
  3. I removed all concepts that were not strictly necessary to understand the work. Expressions such as “quasi-experimental”, “randomized controlled trials”, all the protocols (PRISMA, Cochrane’s, MINORS), etc. are not in the new version. Those concepts that are important (internal validity, efficacy/efficiency, groups of control…) are now defined in every section that they appear.
  4. MDPI editing services reviewed English and style.

I think that the new version of the manuscript is better than the first one. It is shorter, more direct, and easier to read. I hope that changes helped.

Best regards,

Reviewer 3 Report

Many thanks for the opportunity to read and comment on your interesting paper which offers useful insights into this emerging area of interest internationally. I have two main comments:

  • You have mentioned some information on the meta analysis papers which you included in your review however no details of the papers and the specific searches dates etc that you undertook are included this is essential to underpin your recommendations. In addition you need to use headings in order to guide the reader through what you found in your review. Your title suggests a discussion paper yet you claim to have carried out a review and come to conclusions from that you need to be clearer about what you did. 
  • In lines 279-289 and 336-341 you mention that animal assisted practice can only be as an adjunct to an existing discipline, this rather limits the potential for this area of practice and I would not say that your review findings underpin this statement adequately. It is important that your review is clear in terms of what papers it looked at and the findings you made from that and that you are not appearing to try and control the development of this area without adequate review findings to underpin your conclusions. This is not scientific as an approach and framing a review in order to make a point is an opinion paper one which needs to be reviewed in a different way and the paper would need to be structured differently. Currently the title and summary and abstract for this paper are not clear. You need to either write an opinion paper as per your title or undertake a review of meta analysis and write that as a paper. Currently you are claiming to have done a review upon which the reader assumes you are basing your assertions, however it is not clear from the paper what the above outlined assertion is based upon. Is it your opinion, then this is an opinion paper and would need to be written as such.

Author Response

With kind regards, 

Round 2

Reviewer 1 Report

2nd Review, “Current Status of Animal-Assisted Interventions in the Scientific Literature: A Critical Comment on its Internal Validity

I appreciate the extensive work that the author has put into responding to reviewers’ comments in such a short period of time. The revised manuscript also makes clearer the fact that said manuscript is intended as a position or opinion paper, rather than an empirical review of the literature or of meta-analyses of that literature. That said, some problems in the current draft still remain for me.

English language in the manuscript is considerably improved, though some faults still remain. For example, even in the title: the pronoun “its” is singular, but it is being used here to refer to a plural noun (“animal-assisted interventions”) and therefore should be plural. Specifically, the title should read:

“Current Status of Animal-Assisted Interventions in the Scientific Literature: A Critical Comment on Their Internal Validity” (emphasis mine). Same problem in line 18: it should read “…anthrozoological—aspects of AAIs; to consider AAIs as a modality of well-known interventions, avoiding their representations as “alternative,” “new,” or “groundbreaking;”…” (Emphasis mine to make clear where grammatical changes are required).

Line 48: should be “how well they work” rather than “how good they work.”

Such grammatical challenges in regard to the use of English remain (though greatly improved from the initial draft) throughout the paper.

But more substantially, the paper suffers from the lack of a clear, logical, statement about what the author intends to do and how he intends to do it, followed by an equally clear and meticulous abiding by that clearly articulated protocol. And the author’s own use of the term “internal validity” remains “messy,” for lack of a better word.

For example, it remains unclear to the reader whether the critiques being made in this manuscript are about primary peer-reviewed articles published that aim to report data pertaining to the efficacy of AAI in a specific circumstance, or to meta-analyses of that literature, or both. The title implies that the critique is about internal validity in empirical studies of AAI, but the bulk of the body of the manuscript suggests that the critique is actually about internal validity in various meta-analyses of AAI efficacy. Which is it? In many places in the current manuscript, exactly what is being referred to in phrases like “these studies” is ambiguous—is the author talking about meta-analyses? Or the empirical studies included in meta-analyses? Or…..what? Another example: Line 78—what is the target of this charge regarding a “lack of internal validity?” Is this about empirical studies of AAI? Or of the meta-analyses of those studies? Or both? Or…what?

In some parts of the manuscript, the author seems confused himself about the concept of “internal validity.” In Line 52, the author states that (internal validity is about whether or not) “studies measure what they intend to measure.” This is indeed an accurate description of internal validity. However, in the clause immediately preceding this one, the author misrepresents internal validity. It does NOT refer (as is written in lines 51-52) to the “accuracy and reliability” of a study; reliability is a completely different construct, referring to the ability of a measure to obtain the same results when applied to the same individuals multiple times. I could, for example, develop a measure of intelligence that was based on adult shoe size. That measure would be pretty reliable, as shoe size does not change much in adults. However, the validity of such a measure is a different thing entirely. Validity is a construct pertaining to how well (if at all) a given measure actually measures what it claims to measure. Since the construct “efficacy” as applied to the outcome of the application of a proposed therapeutic manipulation is in itself a “fuzzy” construct about which there can be a great deal of argument, the validity of AAI efficacy as an overarching theme for critique in this circumstance is reasonable; however, it is not at all clear that this IS the theme of the current manuscript.

Internal validity is yet again a somewhat different construct from validity, referring to how well the empirical design of a specific study actually demonstrates the cause-and-effect relationship between two variables (in this case, between treatment and outcome—that is, between AAIs and patient improvements).

I think the author is trying to get at this point when he argues in different parts of the manuscript about the lack of a clear theoretical basis for suspecting that such a causal relationship should exist between AAI and a particular therapeutic outcome; the lack of such a theoretical platform makes generating testable hypotheses with clear “if, then” structures that allow for causal deduction based upon analysis of results is indeed a problem. But other than asserting that it is so, the author does not spend much time if any elaborating precisely on why this is so, nor on making the reader understand clearly exactly how this is so.

In other places, the author appears to be using the construct of internal validity to refer to existing meta-analyses of literature on AAIs. For example, he details the variability between such studies in how they chose articles to evaluate. However, as the construct of internal validity is never accurately described in the manuscript, nor are any principles that might be utilized to systematically evaluate it in any given study laid out for consideration, it is impossible again to see clearly how the critiques offered against other meta-analyses of this literature are supported. Secondly, the author himself does a terrible job of describing the rationale and protocol for study selection in his OWN (however informal) “meta-analysis” of existing meta-analyses!

At yet another level, the author implies in the title of the manuscript that it will be the construct of AAIs themselves whose internal validity he will address. But that also does not really happen in this manuscript, nor is it by any means the clearly articulated main point in terms of its follow-through in the manuscript’s main content.

The points that I think the author is trying to make about the current state of the field of AAI are good and important ones, but it is not clear how they are points that others have not already made before, and in this particular case, they are not sufficiently and clearly articulated and documented to make the manuscript in its current state of much additional value to existing literature.

Author Response

Please see atachment.

Best regards, 

Reviewer 2 Report

This manuscript is a commentary on the problems with the present state of research on animal-assisted interventions.  The aims are to examine “current threats that AAIs face,” and discuss the guidelines to “overcome these risks.”  However, the manuscript is an odd combination of a critique of AAI research methodology and commentary on problems with published AAI meta-analyses.

The problems with AAI research methodology are well-known (i.e., lack of control groups, low sample sizes, lack of specificity of treatment procedures, etc.).  Indeed, nearly every systematic review and meta-analysis on the topic ends with a call for more rigorous studies. However, there is a need for a critical examination of AAI meta-analyses

In my view, the manuscript lacked a clear focus and needs a better organization.  A reiteration of threats to the internal validity of AAI research is not needed.  A thorough examination of the state of systematic reviews/meta-analyses of AAI, however, would be of value. Presently, these vary greatly in quality, and some reviews on the same topic using the same studies have come to quite different conclusions. 

Such a project should be put in the context of concerns about the validity of meta-analyses. (See, for example, Greco, et al. 2013 and Biondi-Zoccai, et al. 2011.) It would also involve considerable work as there have been at least 34 meta-analyses and systematic reviews on AAI in the last 10 years alone.

There are lots of errors in the reference section.  (See, for example, reference 11.)

Author Response

Please see atachment.

Best regards, 
